# The CLEAR Benchmark:
# Continual LEArning on Real-World Imagery

**Zhiqiu Lin**[1]    **Jia Shi**[1]    **Deepak Pathak**[1*]    **Deva Ramanan**[1,2*]

[1]Carnegie Mellon University    [2]Argo AI

## Abstract

Continual learning (CL) is widely regarded as crucial challenge for lifelong AI. However, existing CL benchmarks, e.g. Permuted-MNIST and Split-CIFAR, make use of artificial temporal variation and do not align with or generalize to the real-world. In this paper, we introduce CLEAR, the first continual image classification benchmark dataset with a natural *temporal evolution of visual concepts* in the real world that spans a *decade* (2004-2014). We build CLEAR from existing large-scale image collections (YFCC100M) through a novel and scalable low-cost approach to *visio-linguistic dataset curation*. Our pipeline makes use of pretrained vision-language models (e.g. CLIP) to interactively build labeled datasets, which are further validated with crowd-sourcing to remove errors and even inappropriate images (hidden in original YFCC100M). The major strength of CLEAR over prior CL benchmarks is the smooth temporal evolution of visual concepts with real-world imagery, including both high-quality labeled data along with abundant unlabeled samples per time period for continual semi-supervised learning. We find that a simple unsupervised pre-training step can already boost state-of-the-art CL algorithms that only utilize fully-supervised data. Our analysis also reveals that mainstream CL evaluation protocols that train and test on iid data artificially inflate performance of CL system. To address this, we propose novel "streaming" protocols for CL that always test on the (near) future. Interestingly, streaming protocols (a) can simplify dataset curation since today's testset can be repurposed for tomorrow's trainset and (b) can produce more generalizable models with more accurate estimates of performance since *all* labeled data from each time-period is used for *both* training and testing (unlike classic iid train-test splits).

## 1   Introduction

Web-scale image recognition datasets such as ImageNet [50] and MS-COCO [35] revolutionized the field of machine learning and computer vision by becoming touchstones for the modern algorithms [13, 21, 24]. These benchmarks are designed to solve a *stationary* task where the distribution of underlying visual concepts is assumed to be same during train and test. However, in reality, most ML models have to cope with a *dynamic* environment as the world is changing over time. Figure 1 shows web-scale visual concepts that have naturally evolved over time in the last couple of decades. Although such dynamic behaviors are readily prevalent in web image collections [27], recent learning benchmarks as well as algorithms fail to recognize the temporally dynamic nature of real-world data.

That said, there exists a tremendous body of work on continual/lifelong learning, with the aim of developing ML models that can adapt to *dynamic* environments, e.g., non-iid data streams. Many algorithms [18, 28, 30, 34, 41, 44, 58] have been purposed to combat the well-known failure mode of catastrophic forgetting [17, 42, 54]. More recently, new algorithms and metrics [12, 38] have been introduced to explore other aspects of CL beyond learning-without-forgetting, such as, the ability to transfer knowledge across tasks [12, 38] as well as to minimize model size and sample storage [12]. However, all of the above works were evaluated on datasets and benchmarks with synthetic temporal

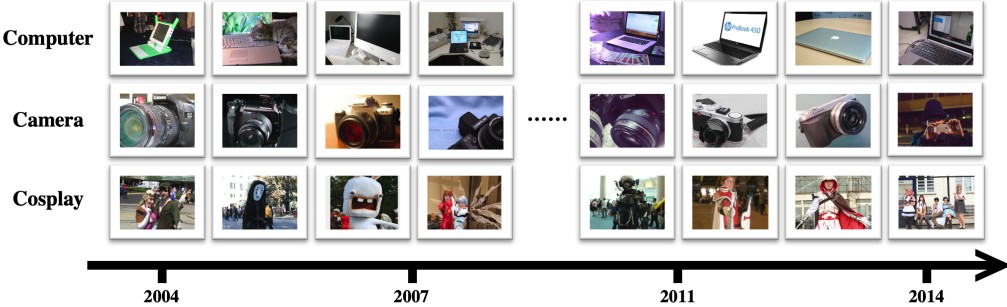

Figure 1: **Temporal evolution of visual concepts in Internet images.** We show the evolution of three concepts (`computer`, `camera`, and `cosplay`) from the Flickr YFCC100M with timestamps spanning 2004 to 2014. The industry advanced rapidly over this decade from Canon EOS 30D (2006) to Canon EOS 6D (2013), and from Apple Powerbook G4 (2004) to Macbook Pro (2011) with substantial design changes. The definition of visual concepts also expanded, e.g., the common usage of the term `camera` evolved from standalone ones to ones in smartphones (iphone 5 in last image of second row). We see evolution in other visual concepts such as `cosplay` as well: Cosplayers often dress as topical characters of the day. From 2004-2007, popular characters reflect anime such as `Rayman Rabbit` (2006), `Rebuild of Evangelion` (2007), etc. while 2011-2014 includes characters such as `Assassin's Creed II` (2010), `Steins;Gate` (2011) and so on.

variation because it is easier to artificially introduce new tasks at arbitrary timestamps. Such continual datasets tend to contain *abrupt* changes:

- *Pixel permutation* (e.g., Permuted MNIST [18]) fully randomizes the positions of pixels at discrete intervals to form new tasks.
- *Incremental scenarios on static datasets* (e.g., Split-MNIST [58] and Split-CIFAR [29, 47]) split existing datasets designed for "static" evaluation to multiple tasks, each with a random subset of classes.
- *New Instances (NI) learning* (e.g., CORe50 [36]) adds brand new training patterns to existing classes. Yet these new patterns are all artificially designed, e.g., whoever collect the images manually change the illumination and background of the captured objects.
- *Two-task transfer* (e.g., Li and Hoiem [34]) trains a single model on a pair of datasets consecutively, such as ImageNet [50] followed by Pascal VOC [15], with the aim of preserving the performance on the first dataset while training on the second one.

**Why are abrupt changes undesirable?** Simulated continual data with abrupt changes is not only unnatural, but may make the problem harder than need be. The real world tends to exhibit *smooth* evolution, which may enable out-of-distribution generalization in biological entities. Indeed, synthetic benchmarks have been criticized [5, 16, 36] for their artificial abruptness because (a) knowledge accumulated on past tasks cannot generalize to future tasks and (b) degenerate solutions such as GDumb [44] that train "from scratch" on new tasks still perform quite well. Aware of these criticisms, we propose this new CL benchmark to promote natural and smooth distribution shifts, and experiments in this paper confirm that GDumb [44] falls short compared with other baselines.

**The CLEAR Benchmark.** In this work, we propose the CLEAR Benchmark for studying Continual LEArning on Real-World Imagery. To our knowledge, CLEAR is the first continual image recognition benchmark based on the natural temporal evolution of visual concepts of Internet images. We curate CLEAR from the largest available public image collection (YFCC100M [52]) and use the timestamps of images to sort them into a temporal stream spanning from 2004 to 2014. We split an iid subset of the stream (around 7.8M images) into 11 equal-sized "buckets" with 700K images each. The labeled portion of CLEAR is designed to be similar in scale to popular ML benchmarks such as CIFAR [29]. For each of the bucket $1^{st}$ to $10^{th}$, we curate a small labeled subset consisting of 11 temporally dynamic classes (10 illustrative classes such as `computer`, `cosplay`, etc. plus an $11^{th}$ `background` class) with 300 labeled images per class. Besides high-quality labeled data, the rest of the images per bucket in CLEAR can be viewed as large-scale unlabeled data, which we hope will spur future research on continual semi-supervised learning [20, 45, 51, 57].

**A low-cost and scalable dataset curation pipeline.** However, constructing such a benchmark a natural continuity is non-trivial at a web-scale, e.g., downloading all images of YFCC100M [52] already takes weeks. To efficiently annotate CLEAR, we propose a novel *visio-linguisitic dataset curation* method. The key is to make use of recent vision-language models (CLIP [46]) with text-

prompt engineering followed by crowdsourced quality assurance (i.e. MTurk). Our semi-automatic pipeline makes it possible for researchers to efficiently curate future datasets out of massive image collections such as YFCC100M. With this pipeline, we curate CLEAR with merely one day of engineering effort, and believe it can be easily scaled up to orders-of-magnitude more data. We will host CLEAR as well as its future and larger versions on https://clear-benchmark.github.io.

**A realistic "streaming" evaluation protocol for CL.** Realistic temporal evolution encourages smooth transitions while suggestive of a more natural *streaming* evaluation protocol for CL, inspired by evaluation protocols for online learning [4, 14]. In essence, one deploys a model trained with present data at some point in the future. Interestingly, traditional CL evaluation protocols train and test on iid data buckets, failing to model this domain gap. We conduct extensive baseline experiments on CLEAR by simply fixing the label space to be the same 11 classes across time (i.e., the incremental domain learning setup [22]). Preliminary results confirm that mainstream (train-test) "iid" evaluation protocols artificially inflate performance of CL algorithms. Our streaming protocols can (a) simplify dataset curation since today's testset can be repurposed for tomorrow's trainset and (b) produce more generalizable models with more accurate estimates of performance since all labeled data from each time period is used for both training and testing.

**Large-scale unlabeled data boosts CL performances.** Moreover, we find that unsupervised pre-training (MoCo V2) on only the first bucket $0^{th}$ (700K unlabeled images) of CLEAR already boosts the performance of all state-of-the-art CL techniques that make use of only labeled data. In particular, training a linear layer to classify these MoCo extracted features surpasses all popular CL methods by a large margin. This suggests that future works on real-world CL should embrace large-scale unlabeled data to maximize performances.

## 2 Background and Related Work

### 2.1 Existing CL Datasets and Benchmarks

Most established works on CL focus on overcoming catastrophic forgetting [7, 18, 28, 30, 31, 33, 41, 44, 55, 58] through replay-based, regularization-based, distillation-based, and architecture-based methods. We refer readers to [10, 43, 44] for more surveys and overviews. However, these algorithms are all evaluated on CL benchmarks with synthetic temporal evolution like Permuted-MNIST [18], CoRE50 [36], and other incremental learning scenarios [10, 29, 40, 58]. The growing field of continual and lifelong learning is in dire need of more practical benchmarks. One notable exception to contrived incremental scenarios is the recent work of Hu et. al [23], who assemble a CL dataset of Tweet messages that naturally evolve over time. Cai et. [4] concurrently introduce a CL dataset also derived from YFCC100M [52], but formulate the task as geolocalization (making use of readily-available geostamps). We focus on the task of image classification, which is arguably more mainstream but requires presumably costly dataset curation.

### 2.2 Continual Learning Settings

Existing works in CL have proposed a variety of CL settings. In this section we explain some of the major CL settings that CLEAR adopts and refer readers to [1, 22, 44, 53, 59] for more thorough discussion of different variants of CL setups.

**Task-based sequential learning:** In most CL works, a sequence of distinct tasks with clear task boundaries is given, and the tasks are iterated in a sequential fashion. This predominant CL paradigm is called *task-based sequential learning* [44] (or "boundary-aware" CL [33]). This setting is easy to set up and benchmark (e.g., Split-MNIST) and has spurred many classic CL algorithms such as EWC [28] and SI [58] that heavily rely on task boundaries in order to know when to perform core model updates (usually at the end of each task), such as knowledge consolidation. In this paper, we also adopt task-based sequential learning with a sequence of (same) 11-way classification tasks by splitting the temporal stream into 11 buckets, each consisting of a labeled subset for training and evaluation. However, it could be argued that in real-world, the model will not be informed about the task boundary (also called boundary-agnostic [33], task-free[1], or task-agnostic CL [59]). Such boundary-agnostic settings have been explored in recent works [1, 4, 23, 59], in which a non-iid data stream continuously spits out new samples without a notion of task switch. In this paper, we still assume a task-based sequential learning setting to ease benchmark design, but future works could

adapt CLEAR to boundary-agnostic or task-free CL by processing data in an online streaming fashion using timestamps of CLEAR images. Moreover, it should be noted that in task-based sequential learning, only the current task data is available at each timestamp (excluding past data), though replay-based methods [6, 44] can still use an external replay buffer to store past data for rehearsal.

**Locally-iid assumption:** Almost all prior CL works on task-based sequential learning setup adopt a naive "locally-iid" assumption, under which each task's data is from an iid distribution [38]. In fact, mainstream CL evaluation protocols that sample train and test data from the same iid distribution make sense only when this iid assumption holds. In this work, we propose novel "streaming" evaluation protocols that test the current model on the data of next task, which does not implicitly assume that each task has its own iid distribution. Note that in order to align with past works, we report comprehensive results under both "iid" and "streaming" protocols in this paper.

**Incremental task/domain/class learning:** [22, 53] categorize existing CL setups for task-based sequential learning into three incremental learning scenarios. In incremental task learning, the task identity of each test sample is *known*; such a-prior knowledge could be exploited to ease algorithmic design, such as training separate classification heads for each task and using task identities of each test sample to determine which head to use. Incremental domain and class learning setups are more challenging since the task identity is *unknown* during test time. We adopt incremental domain learning in this work by fixing the label space for all tasks (same 11-way classification) while the input distribution is changing over time. CLEAR can be adapted for incremental class learning in future by assigning different classes to different tasks, making the label space grow over time.

**Online v.s. offline continual learning:** To encourage quick model adaption in CL, some prior works on task-based sequential learning require each sample to be used just once for model update. This setting is called online CL [40, 44] since it mimics an online stream of data, in which samples are spitted out one by one and cannot be revisited except when it is stored in a memory buffer (usually of limited size). On the other hand, offline CL assumes all samples in current task (plus the ones in buffer) can be revisited without constraint. However, online CL does not imply quick model adaption unless one carefully measures the total resource consumption as argued in [44]. For example, GEM [38], a well-known online CL method, uses each sample only once but solves an expensive quadratic program per model update. Therefore, we adopt offline CL in this work, but we allocate roughly the same resources (e.g., buffer size and training time) for each baseline algorithm for fair comparison. Interestingly, [4] adopts a learning setup similar to our "streaming protocol" but name it as *online continual learning*, which clashes with previous definitions of online CL [40, 44]. In this paper, we use the term "streaming" to avoid a notational clash.

## 2.3 Building Blocks for CLEAR

**YFCC100M:** We build CLEAR from YFCC100M [52], consisting of media artifacts uploaded to `www.flickr.com` from 2004 to 2014. YFCC100M's massive scale (around 100 million images and videos) and the wealth of metadata (timestamps for capture and upload date, GPS, user tag, image description, camera specs, *etc.*) makes it one of the largest and richest publicly-available image datasets. However, YFCC100M is cumbersome to work with as downloading the data can already take months and user-uploaded hashtags and descriptions can be extremely noisy and even irrelevant [3, 25, 31, 39], arguably limiting its impact compared to relatively smaller, yet high-quality curated data like ImageNet [50] and MS-COCO [35]. Nonetheless, YFCC100M consists of real-world and more complex imagery unlike other popular datasets favoring only centered objects or visual elements, e.g., MNIST [11], CIFAR [29], and ImageNet [50]. We believe it is more practical to develop algorithms and models on real-world data such as YFCC100M.

**CLIP:** CLIP [46] is a vision-language model that learns to associate texts and images by training on a massive dataset of over 400M image-text pairs. We use CLIP to automatically retrieve subsets of YFCC100M most relevant to particular visual concepts (Figure 2).

## 3 CLEAR: Dataset Design and Curation

We describe how we curate CLEAR from YFCC100M [52] and CLIP [46]. Because YFCC100M is too large (the metadata files already exceed 40 GBs) to gather and annotate, we believe our dataset curation procedure can ease future benchmark creation, as this pipeline can be managed without

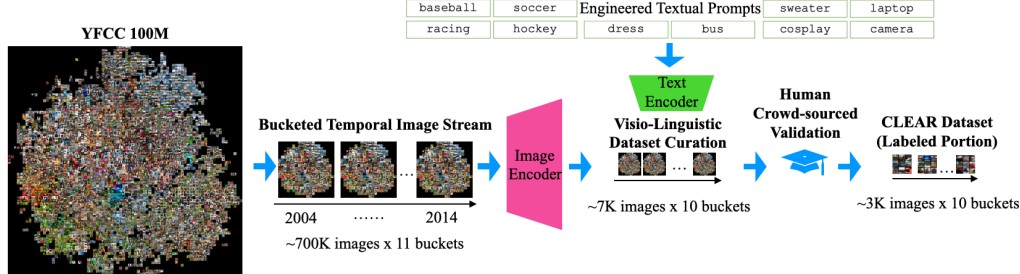

Figure 2: **Visio-Linguistic Dataset Curation.** We download a random subset of 7.8M images and their associated metadata from YFCC100M. We use upload timestamps to reconstruct a temporal stream, splitting it into 11 time-indexed buckets of roughly 700K images each. Given a list of text queries (found by text-prompt engineering in order to effectively retrieve the visual concepts of interest), we use CLIP [46] to extract their respective L2-normalized query features, ranking each image by the cosine similarity of its image feature to each query. We assign the top-ranked (0.7K out of 700K) images of each bucket to the query, removing ambiguous images that rank high across multiple queries. We also include a *background* class with images that rank low across the queries (details in Sec 1. of supplement). We then use human crowd-sourcing (MTurk) to remove misclassified and inappropriate images from the CLIP-retrieved images. As a result, CLEAR contains 3.3K high-quality labeled images for 10 buckets (excluding bucket $0^{th}$ for unsupervised pre-training only). Our dataset curation pipeline reduces the annotation cost by $99\%$.

massive infrastructure or engineering efforts from big organizations. We summarize the entire pipeline in Fig. 2.

**Concept selection:** We select temporally dynamic visual concepts from following super-categories:

- **Trends and Fashion**: People's aesthetics and interests shift over time. A fashionable dress in 2004 may be deemed outdated in 2014. Similarly, the clothing style for popular music performers were also changing, e.g. punk style in 1970s and Kpop idols in 2010s.
- **Consumer products**: Industry is constantly producing new commercial products to meet shifting consumer needs, e.g., models of vehicles, cameras, laptops, and cellphones change routinely.
- **Social Events**: Social/multimedia events are often dated and evolving, e.g., the FIFA world cup features different themes every 4 years. Cosplayers tend to dress as topical characters of-the-day.

We choose 10 dynamic visual concepts that span the above super-categories: `computer`, `camera`, `bus`, `sweater`, `dress`, `racing`, `hockey`, `cosplay`, `baseball`, and `soccer`. Please refer to supplement for a detailed discussion of these visual concepts and see examples in Fig. 1.

**Stream recreation:** To recover the temporal evolution of visual concepts from YFCC100M, we sort images of YFCC100M by their uploaded dates to recreate the temporal stream of Flickr images from 2004 to 2014. In the interest of time, we only downloaded the first 7.8M images offered by their metadata files which are a random subset of YFCC100 images. We then chunk the uploading stream to 11 buckets of 700K images each, indexed from $0^{th}$ to $10^{th}$. We gather a small class-balanced set of 3.3K labeled data for buckets $1^{st}$ to $10^{th}$ with the 10 dynamic visual concepts plus a $11^{st}$ `background` class. The $0^{th}$ bucket does not have a labeled set because we only use it to pretrain an unsupervised MoCo V2 model [9] for feature extraction in subsequent buckets (Sec. 5).

**Visio-linguistic curation:** We work on each of the 10 buckets of the temporal image stream independently to gather 10 labeled subsets consisting of the above dynamic visual concepts. Since it would be too costly to manually label all 700K images per bucket, we use CLIP [46] to facilitate the annotation process. Given a pair of (image, text query), CLIP first encodes both image and query to two normalized features of same dimension (1024), and then performs a dot product between the two features to calculate a cosine similarity score. It has been shown [46] that the higher the score is, the more aligned the image content is to the query content. We then retrieve images with top $0.1\%$ cosine similarity scores with respect to each given query in order to filter out most irrelevant images. As suggested in [46], one can refine textual queries to better capture a visual concept. In particular, we found it useful to enumerate *sub*category queries (e.g., use `laptop` queries to retrieve `computer` images). Please refer to supplement (Sec. 1) for more details. Finally, to assemble `background` images, we construct a set of images that are low-scoring across all queries. Details about how `background` class is constructed are in Sec. 1 of supplement.

**Crowd-sourced validation (Mturk):** We find that CLIP still produces a roughly 20% misclassification rate (though this varies by class). We use Amazon Mturk for crowd-sourced validation to filter

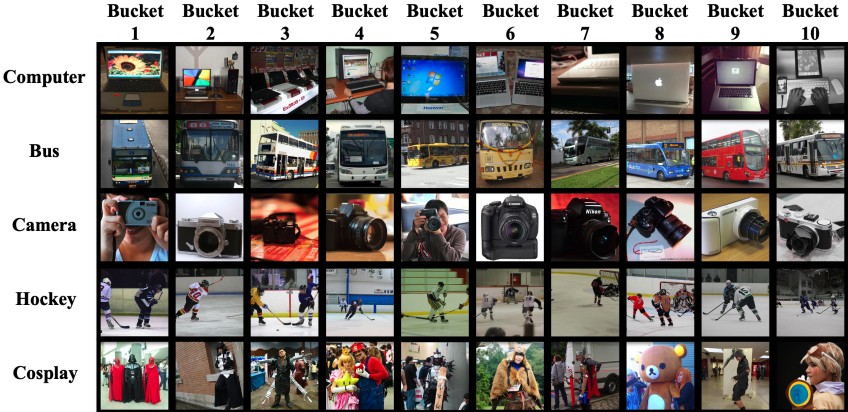

Figure 3: **Example images from CLEAR.** For each bucket (per column), we show a random sample from 5 of the classes (`computer`, `bus`, `camera`, `hockey`, `cosplay`) in CLEAR.

out misclassified images. Each image in our dataset is verified by at least 3 workers. Additionally, we also use MTurk workers to mark any images with inappropriate content. Our pipeline successfully surfaced pornographic images contained in YCFF100M (that we have removed from CLEAR and subsequently reported to the original benchmark curators). Sec. 1 of supplement also shows how we design the MTurk user interface and compose the worker results. Examples of verified images in our final dataset can be found in Fig. 3 and Sec. 1 of supplement.

## 4 Evaluation Protocols for Continual Learning

When presenting results on CLEAR, we make use of standard CL evaluation protocols to align with past works relying heavily on "locally-iid" assumption; many of them focus on an "iid" evaluation on test samples drawn from the same iid distribution of training samples. Instead, we advocate on a "streaming" perspective that evaluates on test data from future, motivated by real-world deployments that notoriously struggle with domain shifts between future test data and past train data. Moreover, such a streaming perspective *simplifies* dataset curation since *today's testset can be repurposed for tomorrow's trainset*. We will show that this translates to both improved models and more robust estimates of performance, since instead of making a classic 70/30% train-test split, we can use *all* labeled data of a bucket for both training and testing. Before we formalize the streaming evaluation protocol, we first review mainstream "iid" evaluation protocols for CL.

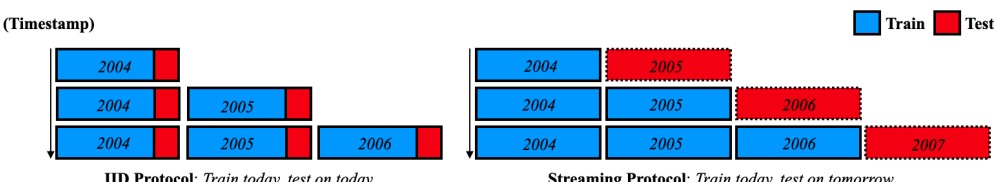

Figure 4: **IID vs Streaming Protocols for CL.** Traditional CL protocols (**left**) split incoming data buckets into a train/test split, typically 70/30%. However, this may overestimate performance since the train and test data are drawn from the same iid distribution, ignoring the train-test domain gap. We advocate a streaming protocol (**right**) where one must always evaluate on near-future data. This allows us to repurpose today's testset as tomorrow's trainset, increasing the total amount of recent data available for both training and testing. Note that the streaming protocol naturally allows for asynchronous training and testing; by the end of year 2006, one can train a model on data up to 2006, but needs additional data from 2007 to test it.

### 4.1 Review of IID Protocols

Following the "locally-iid" assumption, we have $N$ timestamps with a sequence of unknown distributions $\mathcal{D} = \{D_1, D_2, \cdots, D_N\}$ with $D_i = \mathbf{X}_i \times \mathbf{Y}$, where $\mathbf{X}_i \subset \mathbf{X}$ is the input space at timestamp $i$ and $\mathbf{Y}$ is the label space. For CLEAR, $\mathbf{X}_i$ is the image distribution from which bucket $i$ is sampled and $\mathbf{Y}$ contains the 11 dynamic classes. The standard CL evaluation protocols then make a train-test **iid** assumption: Each task consists of a training set $Tr_i$ and a test set $Te_i$ sampled from the same distribution $D_i$ at each timestamp. A learner then proceeds by sequentially fitting $N$ predictor

functions $\{h_1, h_2, \cdots, h_N\}$ on the $N$ training sets. Each predictor function $h_i : \mathbf{X} \to \mathbf{Y}$ will be evaluated on all $N$ test sets to generate an *accuracy matrix* $\mathcal{R} \in [0, 1]^{N \times N}$: $\mathcal{R}_{i,j}$ is defined to be the test accuracy of $h_i$ on $Te_j$. The above formulation can be extended to a label space that grows over time to accommodate new classes (i.e., incremental class learning). However, for simplicity, we focus on "incremental domain learning" (all tasks share a fixed output space while only the input domain is changing), leaving "incremental class learning" (output space is changing as well) on CLEAR as future work. Note that these taxonomies are summarized in [22].

Standard iid evaluation protocols of CL algorithms mostly adopt the following 3 metrics, which can be readily calculated from accuracy matrix $\mathcal{R}$:

1. **In-domain Accuracy** (termed Average Accuracy in [12, 38]) measures the test accuracy on the current task immediately after training on it (averaged over all timestamps). This can calculated as the average of the diagonal entries of $\mathcal{R}$.
2. **Backward Transfer** measures the performance of previous tasks (i.e., learning without forgetting) by averaging lower triangular entries of $\mathcal{R}$.
3. **Forward Transfer** measures performance of future tasks (i.e., generalizing to future) by averaging upper triangular entries of $\mathcal{R}$.

We refer readers to Sec. 3 of supplement for equations we use to calculate these metrics.

### 4.2 Our Streaming Protocol

In real-world deployment scenarios, one must train on today's data and test on tomorrow's, introducing an undeniable domain shift, as demonstrated in Fig. 4.

A more realistic CL scenario is therefore to evaluate on the immediate *next* time period. Formally, we define a "streaming" evaluation protocol for CL. Given $N$ timestamps, we have a stream of data $S = \{S_1, S_2, \cdots, S_N\}$; $S_i$ could be a single sample or a bucket of samples, drawn from a non-stationary distribution $D_i = \mathbf{X_i} \times \mathbf{Y}$. In CLEAR, the stream $S$ is the 10 buckets of labeled data (bucket $1^{st}$ to $10^{th}$). A learner sequentially fits $N$ predictor functions $\{h_1, h_2, \cdots, h_N\}$ on $S_1$ to $S_N$. We call this protocol "streaming" because after training on $S_i$, we evaluate $h_i$ on $S_{i+1}$, which are samples of the next timestamp. Once evaluation is done, $S_{i+1}$ can be repurposed as the new trainset for fitting the next predictor $h_{i+1}$. This is similar to online learning protocols [4, 14], because the current predictor $h_i$ will first make prediction on $S_{i+1}$, and the environment then reveals ground-truth labels of $S_{i+1}$ for evaluation and model update.

Under streaming protocol, accuracy matrix $\mathcal{R}$ can be defined: $\mathcal{R}_{i,j}$ is the accuracy of $h_i$ on $S_j$. We then define *"next-domain"* accuracy to be the average accuracy of current predictor evaluated on the data of next timestamp (i.e., accuracy of $h_i$ on $S_{i+1}$). This translates to the average of *superdiagonal* of $\mathcal{R}$:

$$\textbf{Next-domain Accuracy} = \frac{\sum_{i=1}^{N-1} \mathcal{R}_{i,i+1}}{N-1} \tag{1}$$

Our streaming protocol makes use of much more data for both testing and training: instead of slicing up data into a 70-30% train-test split, we can use 100% of data from a time period for testing (when first encountered) and 100% of that data for training (after time moves forward). This may result in better models and more accurate (e.g., lower variance) estimates of performance [54]. The price we pay is we can no longer evaluate on historical tasks for "learning-without-forgetting" since there is no longer any held-out test data. But from a truly streaming perspective, evaluating an "outdated" task may not be as relevant as more accurate models and robust performance estimates of the task-at-hand.

We stress that CLEAR can be evaluated under both iid and streaming protocols (of which we are aware). We perform exhaustive experiments on both protocols, but highlight the latter because it has been historically underexplored. Especially, the domain shift between future test data and past training data is usually the bottleneck in real-world continual deployments.

## 5 Approaches

**Fully-supervised baselines:** We evaluate state-of-the-art CL algorithms on CLEAR using implementation from an open-sourced CL library Avalanche [37]. In specific, we test replay-based methods such as **ER** [49], **AGEM** [6], and **GDumb** [44], while allowing them to maintain a sufficiently large

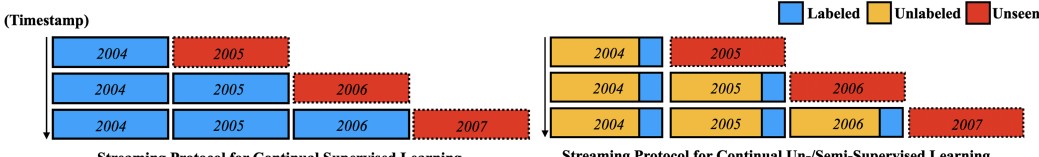

Figure 5: **Streaming Protocols for Continual Supervised vs. Un-/Semi-supervised Learning.** We compare streaming protocols for continual supervised (**left**) and un-/semi-supervised learning (**right**). In real world, most incoming data will not be labeled due to the annotation cost; it is more natural to assume a small labeled subset along with large-scale unlabeled samples per time period. In this work, we achieve great performance boosts by only utilizing unlabeled samples in the first time period (bucket $0^{th}$) for a self-supervised pre-training step. Therefore, we encourage future works to embrace unlabeled samples in later buckets for continual semi-supervised learning.

| Network | Unsup Repr. | Sampling Strategy | Method | IID Protocol | | | | | Streaming Protocol | |
|---|---|---|---|---|---|---|---|---|---|---|
| | | | | In-domain Acc | Next-domain Acc | Acc | BwT | FwT | Next-domain Acc | FwT |
| ResNet18 | - | N/A | EWC [28] | 76.6% ± .2% | 74.3% ± .6% | 76.7% ± .3% | 76.5% ± .4% | 71.1% ± .6% | 77.1% ± .6% | 74.4% ± .6% |
| ResNet18 | - | N/A | SI [58] | 76.0% ± .2% | 73.6% ± .2% | 76.0% ± .5% | 76.0% ± .6% | 71.0% ± .4% | 76.9% ± .2% | 74.3% ± .2% |
| ResNet18 | - | N/A | LwF [31] | 77.8% ± .3% | 75.7% ± .3% | 79.2% ± .3% | 79.6% ± .3% | 72.5% ± .3% | 78.8% ± .2% | 76.1% ± .3% |
| ResNet18 | - | N/A | CWR [36] | 69.5% ± .2% | 67.8% ± .3% | 68.9% ± .3% | 68.8% ± .3% | 66.6% ± .3% | 71.1% ± .4% | 69.9% ± .3% |
| ResNet18 | - | GDumb [44] | GDumb [44] | 66.0% ± .4% | 64.3% ± .5% | 68.4% ± .4% | 68.9% ± .4% | 61.4% ± .5% | 67.4% ± .1% | 64.9% ± .1% |
| ResNet18 | - | ER [49] | ER [49] | 77.3% ± .1% | 75.6% ± .3% | 79.0% ± .1% | 79.3% ± .1% | 72.4% ± .2% | 78.1% ± .2% | 75.8% ± .2% |
| ResNet18 | - | Reservoir | AGEM [6] | 76.2% ± .3% | 73.6% ± .2% | 75.9% ± .2% | 75.9% ± .3% | 70.7% ± .2% | 77.4% ± .2% | 74.5% ± .2% |
| ResNet18 | - | Reservoir | Finetuning | 69.5% ± .3% | 67.7% ± .2% | 70.0% ± .2% | 70.0% ± .1% | 66.5% ± .2% | 71.6% ± .2% | 70.6% ± .2% |
| ResNet18 | - | Biased Reservoir | Finetuning | 75.5% ± .2% | 72.7% ± .3% | 75.7% ± .2% | 75.8% ± .2% | 70.2% ± .2% | 77.2% ± .3% | 74.4% ± .2% |
| Linear | YFCC-B0 | N/A | EWC [28] | 91.6% ± .1% | 90.6% ± .0% | 91.7% ± .0% | 91.7% ± .1% | 88.6% ± .0% | 91.1% ± .0% | 89.3% ± .0% |
| Linear | YFCC-B0 | N/A | SI [58] | 91.7% ± .0% | 90.5% ± .1% | 91.7% ± .0% | 91.7% ± .0% | 88.5% ± .1% | 91.1% ± .0% | 89.2% ± .0% |
| Linear | YFCC-B0 | N/A | LwF [31] | 91.6% ± .0% | 90.7% ± .0% | 92.2% ± .0% | 92.3% ± .0% | 88.7% ± .1% | 91.2% ± .0% | 89.3% ± .0% |
| Linear | YFCC-B0 | N/A | CWR [36] | 90.5% ± .0% | 89.8% ± .0% | 91.3% ± .0% | 91.4% ± .1% | 88.1% ± .1% | 90.4% ± .0% | 88.9% ± .0% |
| Linear | YFCC-B0 | GDumb [44] | GDumb [44] | 85.5% ± .3% | 85.0% ± .3% | 85.5% ± .2% | 85.5% ± .3% | 84.3% ± .3% | 85.8% ± .1% | 85.2% ± .1% |
| Linear | YFCC-B0 | ER [49] | ER [49] | 91.7% ± .0% | 90.9% ± .0% | 92.2% ± .0% | 92.3% ± .0% | 88.9% ± .2% | 91.4% ± .1% | 89.6% ± .1% |
| Linear | YFCC-B0 | Reservoir | AGEM [6] | 91.9% ± .0% | 90.7% ± .1% | 92.1% ± .1% | 92.1% ± .1% | 88.6% ± .1% | 91.4% ± .0% | 89.4% ± .0% |
| Linear | YFCC-B0 | Reservoir | Finetuning | 89.3% ± .0% | 88.7% ± .0% | 90.3% ± .0% | 90.6% ± .0% | 87.2% ± .0% | 89.4% ± .0% | 88.0% ± .1% |
| Linear | YFCC-B0 | Biased Reservoir | Finetuning | 91.6% ± .0% | 90.5% ± .0% | 91.7% ± .0% | 91.7% ± .0% | 88.5% ± .0% | 91.1% ± .0% | 89.2% ± .0% |

Table 1: **Results of baseline CL algorithms on CLEAR.** We evaluated a variety of SOTA algorithms under both **IID** and **Streaming Protocols**. Under the classic **IID Protocol**, **In-domain Acc** (avg of diagonal entries of $\mathcal{R}$) is consistently larger than **Next-domain Acc** (avg of superdiagonal entries of $\mathcal{R}$), indicating that classic (70%-30%) iid train-test construction overestimates performance of real-world CL systems, which must be deployed on future data. Crucially, this drop can be addressed by our **Streaming Protocol**, which trains on all data of the previous bucket (by repurposing yesterday's testset as today's trainset). Moreover, while most prior CL algorithms make use of supervised learning, we find that unsupervised pre-trained representations (YFCC-B0) can boost performances of all baseline algorithms; especially, linear models are far more effective than ResNet18 trained from scratch, even when using naive **Finetuning** strategy with buffer populated with simple reservoir sampling. Note that for replay-based methods, we keep the same buffer size of one bucket of images (2310 for IID and 3300 for streaming protocol).

buffer of one bucket of training images (2310 for iid protocol, 3300 for streaming protocol). We also test **CWR** [36] (architecture-based), **LwF** [31] (distillation-based), and **SI** [41, 58] and **EWC** [28] (both are regularization-based). We include a naive replay-based **Finetuning** strategy, which is to finetune the model solely on the replay buffer (inspired by GDumb [44]), while exploring variants of reservoir sampling strategy to populate the buffer. For all these baseline methods, we use SGD with momentum to train ResNet18 [21] initialized from scratch and report all hyperparameters in Sec. 6 of supplement. Note that some of these approaches such as AGEM can be used for online CL [40] by visiting each sample just once, but in this work we assume the most relaxed offline CL setup so that all samples in the current bucket can be revisited without constraint.

**Unsupervised pre-training:** Since CLEAR comes with abundant unlabeled samples (700K images per bucket), one may also explore continual semi-supervised or unsupervised learning, as suggested in Fig. 5. As a preliminary experiment, we perform a simple unsupervised pre-training step by self-supervised learning on YFCC100M data collected prior to bucket $1^{st}$. Specifically, we learn an unsupervised representation model (Moco V2 [9]) with ResNet50 backbone on bucket $0^{th}$ of 700K images. We release both our pre-trained MoCo V2 model and self-supervised features (termed as **YFCC-B0**) associated with each image. After pre-training the MoCo model, we follow the popular linear evaluation protocol in self-supervised learning literature [9, 19] to classify the extracted YFCC-B0 features via a linear layer. Surprisingly, we observe significant performance boosts across all methods, even with naive **Finetuning** with simple reservoir sampling strategy. Additionally, in Sec. 5 of supplement, we present linear and nonlinear (2 layer MLP) classification results using a variety of other pre-trained feature representations (including ImageNet, CLIP, and other self-supervised methods).

| $\alpha$ for Biased Reservoir | IID Protocol | | | | | Streaming Protocol | |
|---|---|---|---|---|---|---|---|
| | In-domain Acc | Next-domain Acc | Acc | BwT | FwT | Next-domain Acc | FwT |
| $\alpha = 0.5$ | $88.6\% \pm .0\%$ | $88.1\% \pm .0\%$ | $89.8\% \pm .0\%$ | $90.0\% \pm .0\%$ | $86.8\% \pm .0\%$ | $88.8\% \pm .1\%$ | $87.6\% \pm .1\%$ |
| $\alpha = 1.0$ | $89.5\% \pm .0\%$ | $88.8\% \pm .0\%$ | $90.4\% \pm .0\%$ | $90.6\% \pm .0\%$ | $87.4\% \pm .0\%$ | $89.5\% \pm .0\%$ | $88.1\% \pm .1\%$ |
| $\alpha = 2.0$ | $90.7\% \pm .0\%$ | $89.8\% \pm .0\%$ | $91.2\% \pm .0\%$ | $91.4\% \pm .0\%$ | $89.1\% \pm .2\%$ | $90.3\% \pm .1\%$ | $88.8\% \pm .1\%$ |
| $\alpha = 5.0$ | $91.5\% \pm .0\%$ | $90.4\% \pm .0\%$ | $91.7\% \pm .0\%$ | $91.7\% \pm .0\%$ | $88.5\% \pm .0\%$ | $90.9\% \pm .0\%$ | $89.2\% \pm .0\%$ |
| $\alpha = 0.25 * i/k$ | $89.7\% \pm .0\%$ | $88.8\% \pm .0\%$ | $90.5\% \pm .0\%$ | $90.7\% \pm .0\%$ | $87.1\% \pm .0\%$ | $89.7\% \pm .0\%$ | $88.0\% \pm .1\%$ |
| $\alpha = 0.50 * i/k$ | $90.7\% \pm .0\%$ | $89.7\% \pm .0\%$ | $91.2\% \pm .0\%$ | $91.3\% \pm .0\%$ | $87.9\% \pm .0\%$ | $90.4\% \pm .1\%$ | $88.6\% \pm .1\%$ |
| $\alpha = 0.75 * i/k$ | $91.3\% \pm .0\%$ | $90.1\% \pm .0\%$ | $91.6\% \pm .0\%$ | $91.6\% \pm .0\%$ | $88.2\% \pm .1\%$ | $90.8\% \pm .1\%$ | $88.9\% \pm .0\%$ |
| $\alpha = 1.00 * i/k$ | $91.6\% \pm .0\%$ | $90.5\% \pm .0\%$ | $91.7\% \pm .0\%$ | $91.7\% \pm .0\%$ | $88.5\% \pm .0\%$ | $91.1\% \pm .0\%$ | $89.2\% \pm .0\%$ |

Table 2: **Analysis of biased reservoir sampling.** We consider the case where we pretrain MoCo on $0^{th}$ bucket via unsupervised learning and finetune the last linear layer. Table shows different alpha values for biased reservoir sampling algorithms with replay buffer size of one bucket. Note that the higher the alpha values are, more recent samples will be added to the replay buffer (when $\alpha = 1.0$, it is equivalent to naive reservoir sampling). The key takeaway is that when there is a limited memory buffer, we should bias towards storing more recent samples, e.g., when $\alpha = 5.0$ or $\alpha = 0.75 * i/k$, all metrics enjoy a 1% boost compared with naive reservoir sampling.

**Reservoir Sampling:** For naive **Finetuning**, we adopt reservoir sampling [32, 56] that uniformly samples from all data encountered in the stream to populate the replay buffer. Specifically, given a buffer of size $k$, it repeats the following at each timestamp $i$:

1. If the buffer has not reached its maximal capacity, keep adding new sample.
2. If the buffer is full, when a new sample comes, replace a random sample in the buffer with this new sample with probability $\frac{k}{i}$.

Following this procedure, we can ensure a uniform probability that a visited sample is included in the buffer, i.e., at time $i$, any sample in stream has probability $\frac{k}{i}$ to be stored in buffer. Note that this sampling procedure is performed once per incoming bucket in our implementation by simply assuming that all samples in one bucket share the same timestamp $i$.

This uniform sampling procedure has shown to work well in prior works [8, 48] as well as some variants that tackle class-imbalanced data streams [2, 26]. However, in CLEAR, we will show that it is beneficial to simply *bias* the probability by an alpha value $\alpha$ larger than 1.0, i.e., $\alpha * \frac{k}{i}$ such that the sampling procedure favors more recent samples.

**Biased Reservoir Sampling:** In particular, we experiment with two types of alpha:

- **Fixed** $\alpha$. We can select alpha as a constant value, e.g., $\alpha \in \{0.5, 1.0, 2.0, 5.0\}$. Note that $\alpha = 1.0$ is equivalent to unbiased reservoir sampling. Higher $\alpha$ biases the memory towards storing more recent samples (and vice-versa).
- **Dynamic** $\alpha$. The alpha value could also change according to the timestamp $i$. For example, we can have $\alpha \in \{\frac{0.25i}{k}, \frac{0.5i}{k}, \frac{0.75i}{k}, \frac{i}{k}\}$. In this scenario, the probability of replacing an old sample in buffer with a new sample is always a constant, e.g. when $\alpha = \frac{i}{k}$, we always store the new sample with probability $1 = \alpha * \frac{k}{i}$. The latter is equivalent to a first-in first-out (FIFO) priority queue that ensures the $k$ most recent examples remain in memory. Interestingly, many recent works on online CL also adopt FIFO queues [4, 23].

In Sec. 4 of supplement, we provide pseudocode of biased reservoir sampling algorithm.

## 6 Results and Discussion

In this section, we include a summary of the salient conclusions. We run 5 random seeds for each experiment and report mean and std over 5 runs. For the **IID Protocol**, we use 5 random 70-30 train-test splits, reporting both **In-domain Acc** and **Next-domain Acc**, plus three other metrics inspired by prior work [12] including Accuracy (**Acc**), Backward Transfer (**BwT**), and Forward Transfer (**FwT**). For the **Streaming Protocol** (that repurposes previous testsets as trainsets), we report **Next-domain Acc** (Eq.1) and Forward Transfer (**FwT**). Table 1 presents all baseline results.

**In-domain Acc inflates performance**: We first demonstrate that **In-domain Acc** can falsely inflate the performance of CL algorithms that must be deployed on real-world data from the immediate future. Figure 6 includes an accuracy matrix with Linear, YFCC-B0, Bias Reservoir Sampling strategy. Clearly, accuracies on the main diagonal (where train and testsets are drawn from the same iid distribution) are larger than those on the superdiagonal. The accuracy drops on the superdiagonal

(train today, test on tomorrow), and continues to drop as we evaluate further into the future (towards the right of the matrix), suggesting CLEAR contains smooth temporal evolution of data. Crucially, Table 1 shows that this drop can be partially addressed by our **Streaming Protocol** that trains on all 100% of prior data (rather than 70%, as dictated by classic iid protocols).

**Unsupervised pre-training significantly boosts performance**: Without unsupervised pre-training, training ResNet18 on raw RGB images with state-of-the-art fully-supervised CL techniques achieves at best $77.8\%$ **In-domain acc** (under iid protocol) and $78.8\%$ **Next-domain acc** (under streaming protocol) using LwF [31]. However, **Finetuning** a linear layer on top of unsupervised feature representations (YFCC-B0) pre-trained on bucket $0^{th}$ of CLEAR improves performance to $91.9\%$ **In-domain acc** (under iid protocol) and $91.4\%$ **Next-domain acc** (under streaming protocol). This suggests that CLEAR is still challenging without unsupervised pre-training even in the simplest incremental domain learning setup, and future works should embrace unlabeled data for continual semi-supervised learning to maximize performances.

**GDumb falls short compared to other baselines**: GDumb [44] as a degenerate solution is far less competitive on CLEAR, most likely because it trains a network from scratch for each bucket while giving up previously trained models. This suggests that CLEAR has smooth temporal variation, in which case continuous representation learning becomes beneficial. To verify this hypothesis, in Sec. 5 of supplement, we show that it is always better to finetune than to train from scratch per timestamp.

**Biased reservoir towards more recent samples is beneficial**: Biased reservoir sampling that assigns higher sampling probability towards more recent samples combined with naive **Finetuning** improves upon unbiased reservoir and achieves competitive results on CLEAR under both iid and streaming evaluation protocols as in Table 1. We show analysis for different alpha values in Table 2.

|  | Test 1 | Test 2 | Test 3 | Test 4 | Test 5 | Test 6 | Test 7 | Test 8 | Test 9 | Test 10 |
|---|---|---|---|---|---|---|---|---|---|---|
| Train 1 | 90.45% | 89.44% | 87.94% | 88.22% | 87.37% | 85.27% | 86.43% | 85.36% | 85.64% | 82.45% |
| Train 2 | 91.42% | 90.63% | 90.37% | 89.76% | 88.40% | 87.53% | 87.76% | 87.42% | 86.93% | 84.02% |
| Train 3 | 91.39% | 90.93% | 91.48% | 90.79% | 89.42% | 88.52% | 88.48% | 88.02% | 88.22% | 85.61% |
| Train 4 | 91.58% | 91.84% | 91.58% | 91.84% | 90.12% | 88.79% | 89.64% | 88.49% | 89.11% | 86.59% |
| Train 5 | 91.90% | 91.68% | 91.52% | 91.84% | 91.23% | 89.07% | 89.46% | 89.33% | 90.00% | 87.39% |
| Train 6 | 91.45% | 91.82% | 91.66% | 91.21% | 90.81% | 90.99% | 89.88% | 89.86% | 90.06% | 86.83% |
| Train 7 | 93.28% | 92.39% | 92.22% | 91.90% | 91.56% | 90.87% | 91.13% | 90.10% | 90.71% | 87.25% |
| Train 8 | 92.04% | 91.84% | 91.78% | 91.80% | 92.08% | 91.13% | 90.93% | 90.95% | 90.95% | 87.64% |
| Train 9 | 92.59% | 91.86% | 91.96% | 92.30% | 92.36% | 91.01% | 91.35% | 90.67% | 92.16% | 90.54% |
| Train 10 | 91.46% | 91.72% | 91.50% | 92.20% | 92.02% | 91.07% | 91.25% | 90.87% | 91.76% | 92.61% |

Figure 6: **Accuracy matrix under iid protocol with** `Linear, YFCC-B0, Biased Reservoir, Finetuning` **strategy**. We show the accuracy matrix under iid protocol which performs training and testing on the same bucket. The x-axis is the test performance on $Te_1$ to $Te_{10}$ (from left to right), and the y-axis is the training timestamp 1 to 10 (from top to bottom). The main diagonal (**In-domain Acc**) tends to have better performance than the superdiagonal (**Next-domain Acc**) because the former ensures train and test distributions are iid. The further the test bucket is from the current timestamp, the worse the performance, e.g., if we use timestamp $1^{st}$ model to evaluate on $Te_{10}$, the accuracy drops from $90\%$ to $82\%$.

## 7 Conclusion

We present CLEAR, the first benchmark for naturally-evolving continual image classification. We describe a scalable and semi-automatic visio-linguistic approach for (continual) benchmark construction and present a suite of baseline algorithms and analysis. Salient conclusions are as follows (1) Embrace train-test domain shift! Though a widely-held sentiment, it is surprising to see that traditional evaluation protocols of CL still rely on locally iid assumptions. (2) Unsupervised pre-training with simple sampling strategies surpasses state-of-the-art full-supervised CL techniques, and thus future CL benchmarks or algorithms should take unlabeled data into accounts. We plan to host CLEAR benchmark on a public platform (link: https://clear-benchmark.github.io) and maintain leaderboard of different competing approaches to further ensure reproducibility.

**Broader Impacts:** Continual Learning has been heavily studied algorithmically but the evaluation has been plagued with datasets with synthetic changes in the distribution over time. We believe CLEAR is the first step filling the gap between CL benchmarks and real-world deployment. Even though, our dataset is a subset of already existing YFCC100M, we still performed due diligence to ensure that the labeled portion of the dataset is free of inappropriate images. We hope that the real world nature of our benchmark will allow the community to identify biases arising from continual distributions, an under-explored but relevant problem for real-world ML deployments.

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
