# OpenReview forum: "The CLEAR Benchmark: Continual LEArning on Real-World Imagery"
_NeurIPS.cc/2021/Track/Datasets_and_Benchmarks/Round2 — NeurIPS 2021 Datasets and Benchmarks Track (Round 2)_

### Official Review · Reviewer_kspM · 2021-09-16
**New data set variant/subset with interesting ideas that ultimately falls short due to presentation and writing issues**

**Rating:** 5
**Confidence:** 5

**Strengths:**

* The idea of constructing a dataset where concepts naturally evolve over time is certainly interesting. I do agree with the authors that continual learning is in dire need of data sets that are more realistic than permuted MNIST or contrived incremental scenarios.
* The data selection and curation method seems reasonable. I can imagine the combination of CLIP + MTurk resulting in accurate labelling at a much higher throughput.
* An “online” setting, here meant as a streaming setting where data cannot be revisited, can be argued to be intuitively closer to real-world applications. However, also see weaknesses on this part.
* Authors provide a lengthy supplementary material with many nice additional details. The dedicated homepage is also a plus.

**Weaknesses:**

Although the central idea and the proposed dataset is interesting, the way the paper is presently structured comes with some major weaknesses. In my opinion, the paper could be dramatically improved if it were more focused and didn’t attempt to do too many things at once. Right now, unfortunately none of the introduced aspects are very well executed. The paper claims multiple contributions, but I believe some of these are over claimed or partially conflated. Following this point, my biggest concern is the framing of the paper. Initially the paper starts of well with a motivation behind CLEAR, but then somehow diverges into assumptions, desiderata, evaluation, together with potential algorithmic novelties, with a constant connotation that this is the right way to go about and that continual learning up to date is somehow inherently flawed. The dataset itself occupies a rather minor portion of the paper, which is unfortunate. More specifically:

* The authors seem to partially conflate “continual learning”, as a general desideratum, with what is at large a question of domain adaptation. The idea behind continual learning is that there may be various forms of distribution shifts over time.  I find that the authors too readily dismiss that e.g. class incremental scenarios may be valuable. For instance, I do not think that it is unrealistic that I train a system on say distinguishing 1000 flower or insect subspecies, and then later want to add some more to this system. Most importantly, I find that the authors’ evaluation seems to be tailored to fit their narrative. They advocate to measure the system’s worth on the “next-in-line” data, whereas continual learning would care about maximizing performance over all time steps. First, if I mainly care about just the most novel (or future) task, then there is no need for “continual” learning, I can just adapt to my most recent domain unidirectionally. Second, if this is the primary criterion of evaluation, then it is not at all surprising why e.g. favoring more recent instances in the replay buffer (dynamic alpha as called in the authors’ language) would result in better accuracy. This is also evident from table 3. In my opinion, this is a very  strong assumption on how the system gets evaluated. This leads to the third point, why is this a good choice? I would argue that for some objects I no longer require to recognize very old variants (like laptops), but for the majority of real world concepts, this is not the case.
* Extending above point, the precise nature of the distribution shift in the proposed dataset is not clear to me. For computers, I can follow the authors’ narrative that there has been a lot of changes over time that fundamentally change appearance etc. For e.g. Hockey, I am not sure that this is even on a remotely similar level (does Hockey even need continual learning to recognize the concept of a hockey player?). Intuitively, it seems like the major distribution shift here is primarily a consequence of sensor changes.
* Unless I misunderstand something entirely, the statement “CLEAR also comes with abundant unlabeled data per time period” seems to be an attempt to make the contribution look bigger than it actually is. Unless I am mistaken, this data has not been acquired by the authors but simply randomly sampled from an already existing data set. As it is not further labelled, it also does not go through the authors’ proposed pipeline of CLIP + MTurk. The contribution then is basically just sorting + binning the data according to an already existing meta-data time label. This is hardly a contribution, but the authors should feel free to correct me if there is a severe misunderstanding.
* The authors are using a very stochastic way to construct their replay buffer, yet table 2 shows single accuracy values. The authors should provide statistical deviations across experimental repetitions (also because training itself is stochastic anyway).
* The used replay buffer is stated as “size of one bucket of training images”. That seems to be a lot of original data to store, so it is unsurprising that it would work well. It also potentially explains why even a naive strategy as “just uniformly sample a new element and replace an old one with a specific probability” is empirically good enough. The smaller this replay buffer size gets, the more care needs to be attributed to actually maintaining a set of exemplars that spans the observed data distribution.
* At multiple points in the paper the argument is made that YFCC100M is too large to handle and thus a rather small subset is extracted. Although I understand that the final CLEAR dataset is smaller, due to the cost of CLIP + MTurk, YFCC100M is certainly not too large to handle in a modern time. As the authors’ themselves state, “We curated CLEAR with merely one day of engineering effort”. That is very far away from what is typically considered to be a reasonable effort for dataset construction. Even downloading a 10TB dataset is feasible, with costs of 2TB hard drives hardly superseding $100. To clarify, I am not asking the authors to necessarily extend their final CLEAR set, I believe it might already be good enough as an initial dataset, but I would advocate to change this narrative. That said, if the authors could provide an indication for how the performance of algorithms behaves with respect to how many data instances per class+bucket are considered, this would give a very good intuition for future works on whether and how the dataset should be extended.
* The authors’ make a point about their procedure being valuable for future data extraction of different subsets or classes from YFCC100M. However, as the authors state themselves, this dataset seems to no longer be publicly accessible. When I checked the website, even the dataset browser is no longer accessible. Can the authors provide a solution here? Alternatively these statements should be removed.

**Additional Feedback:**

 * As a minor point that the authors can just improve upon easily, instead of writing “see supplementary for details”, there could just be a direct reference to section, figure, table etc. in the appendix.
 * The authors should ideally host their dataset in a form that has a DOI and supports versioning for future changes and reproducibility. Zenodo could be one popular choice.

**Clarity:**

* There is a major concern with the way the words “online” and “offline” are used in the paper. I acknowledge that the authors’ have put these words in quotations on purpose and even initially state that they deviate from the original definition. However,  reading further along the paper this just results in very confusing writing and it is not clear to me why this sort of re-definition needed to happen. In particular, since there is nothing obviously wrong with the initial statement on “online” being viewed as an ongoing stream of data that can only ever be seen once, resulting in immediate updates without revisits.
* Extending above point, what the authors seem to conflate is the fact that most machine learning benchmarks are treated from a closed world perspective, i.e. the captured dataset describes all there is to see. I agree with the authors that the reality is an “open world”, that is, there can be new instances that look entirely different (the camera example is a good one here) and are unknown, and distributions generally drift (additional sensor changes etc.). Rather than making an attempt at re-defining the term online, the authors may want to take a look at open world recognition/learning. This way, the issue of notation clash can also possibly be avoided.


**Correctness:**

* Already some initial statements in the introduction are factually incorrect, like “Dumb implies that CL may even be harmful because from-scratch classifiers tend to perform better by ignoring previous experience” (which they obviously do not if this previous experience is captured through explicitly stored data in a replay buffer).
* The pre-training + unsupervised narrative, the way it is experimentally conducted and compared, seems misleading and unfair to me. E.g. the statement “pre-training step surpassing all popular state-of-the-art CL techniques that only make use of fully supervised data” is a direct consequence of an unfair evaluation.  I can imagine representation transfer being a very interesting aspect to look into, if done and analyzed correctly. The way I understand what the authors do, is that they pre-train a model on the same classes as provided in later evaluation, and then check how much this helps when evaluating these classes under some form of later distribution shift. (This seems less like general continual learning, and more like simple domain adaptation).  It is expected that pre-training will boost your result if the target task is actually the same throughout. But then the authors should also use semi-supervised continual learning techniques (all the generative modeling based approaches come to mind) to contrast.
* Line 215+ states that “offline CL” is equivalent to “unlimited storage” in a sample-storage/size of replay buffer. I do not think this is true. “Offline” evaluation in continual learning just assumes that a “task” can be trained to convergence over the course of multiple epochs (revisiting data) before proceeding to the next “task” (where old task data is then no longer available, in whichever way the task is respectively defined). However, older tasks still get approximated/forgetting gets alleviated through a limited memory replay buffer. I do not see a fundamental difference here.

**Documentation:**

* The documentation is generally nice and the provided dataset sheet and code documentation is very appreciated.
* It is a shame that the authors make a statement that code for evaluation/experiments will be provided conditioned upon paper acceptance. The main purpose of conducting a non-blind review process was to avoid these promises and make the reviewing process more transparent. The authors should ideally have either uploaded their code already, or at least provided a copy in the supplementary upload of the paper submission.

**Ethics:**

* Although the dataset sheet is provided, some parts are skipped and claimed to not apply to the dataset. In particular, the parts on “Does the dataset relate to people” and surrounding questions are falsely answered with “No”. The dataset, as extracted by the authors, does contain multiple categories where real people are featured (baseball, soccer, cosplay). Just because the original dataset has somehow been gathered through Flickr, doesn’t exempt the authors from making a statement here.
* The authors further give an answer to the meta-data question: “Each instance in CLEAR is an RGB image. “ Apart from the question why they would drop some of the originally contained metadata, I am wondering whether this violates the copyright of the initial dataset. YFFC100M clearly states “All the photos and videos provided in the list are licensed under one of the Creative Commons copyright licenses, and as such they can be used for benchmarking purposes as long as the photographer/videographer is credited for the original creation”. Is this actually being done for each extracted image?

**Relation To Prior Work:**

As mentioned above in weaknesses, one of my biggest concerns is the way the paper is framed, also with respect to related work.

* Apart from the fact that neither figure 2 nor table 1 are adequately described in the text, I believe the framing of the related work and the dataset in this context are very problematic. I find table 1 to be very misleading. For a start, it mixes datasets with methods on the y-axis. As an example, “Task-IL” references a methods paper on parameter regularization, dataset pairs one on replay with generative models. These are *algorithmic contributions* and have made *a choice* in their respective evaluation at the time. They are *neither* a dataset contribution, *nor* do they provide a critique or position on what the correct form of evaluation is. As such, their evaluation is a choice made at the time of writing the respective paper. Almost the same problem can be found on the rows of that table. The desiderata as articulated in the referenced work (A-E in the table) are treated as some form of checkmarks for CLEAR, Firehose, permuted MNIST… Desiderata (let’s leave aside whether they are actually desired or not for a moment) such as “shared output head” or “no test-time task labels” are a choice of the way an algorithm is constructed and evaluated, not a property of a dataset. Almost every “contrived” continual learning dataset evaluation on some splits of CIFAR, MNIST, ImageNet has seen methods explore settings where A-E are fulfilled or violated. Again, this is not a property of the dataset, but the way the author’s have dealt with it. The same goes for semi-supervised. I can treat any dataset as unsupervised by simply not using labels, even if they are technically annotated. I believe most of the placed checkmarks are either wrong or very misleading. In addition, neither the terms on the rows, nor the columns are explained properly in the text.
*  A large amount of the citations are incomplete, inconsistent or wrong. To name a few: Shin et al was published at NeurIPS and not arXiv17, Lomonaco et al (reference 33), has 10 authors listed and then et al (either list all or put et al right away) and was not published in CVPR but CVPR-W, gradient episodic memory (reference 34) is not archival but has been published at NeurIPS 17… etc. I suggest the authors go through all of their references again.

**Summary And Contributions:**

The paper introduces the CLEAR recognition benchmark, consisting of 10 temporally evolving concepts across a real world time frame of roughly a decade. The data has been extracted and curated from the large YFCC100M data repository with the help of a multi-modal CLIP model and MTurk crowd-sourcing. Further, the authors propose to change continual learning evaluation scenarios and evaluation protocols and claim the introduction of a “novel” reservoir-sampling algorithm for continual learning.

-----------------------
Post rebuttal update:

The authors have done a great job at addressing many of the concerns and comments raised in my initial review. This includes some clarification with respect to the unsupervised procedure, sharing code and a versioned Zenodo repository, and several acknowledgements of suggestions to improve the writing. I am thus raising my rating. At the same time, I however feel that the amount of suggested changes are too many to give the paper a direct accept rating without being able to check a potentially updated version. There are also some persisting concerns, see response thread below, especially in the misleading presentation of the related works in table 1 and the ethical aspects in the dataset sheet.

---

> ### Author Response · Authors · 2021-09-27
> **Responses to the review (Part 1)**
>
> We appreciate the thorough and thoughtful review. While we concede that CLEAR has a unique point of view, our goal is to ensure that it is adopted by the community and so hope to engage R4 on a constructive discussion for how to do so.
>
> > (1) Is the paper focusing on too many things for now? And is it over claiming or partially conflating the contribution?
>
> * From our perspective, a novel dataset and benchmark should include many things, including a motivation for why it is needed (ie, what “gaps” it fills in the current research landscape), a crisp description of metrics, and a description of baseline architectures. In our case, we also describe the curation process as well, as we believe it can be used for future benchmark construction.Missing any of these aspects would make the story incomplete. As for the dataset itself, we include a thorough discussion on its curation and design in Sec. 1 and 2 in appendix.
> * We happily recognize the importance of prior benchmarks (such as Permuted MNIST)  that gave birth to the numerous state-of-the-art algorithms that combat forgetting issues, but we believe that the time is right to take a step toward real-world practical CL, as formalized in CLEAR. Rather than dismissing prior work, we see CLEAR as a natural evolution of datasets, which historically has been vital to the development of a research field (e.g., Caltech 6 => Caltech101 => Caltech 256 => ImageNet). If you think any of our specific wording or positioning with respect to prior work is unfair or out-of-line, we are happy to revise.
>
> > (2) The paper dismisses class-incremental learning as a valuable CL scenario. And the online accuracy metric concerning only the most novel task is just unidirectional domain adaptation, which may not be a good choice for many applications.
>
> * We agree that class-incremental learning is a valuable CL scenario, and in fact we advocate in the paper (Table 1) that CLEAR can support both domain- and class-incremental learning. Rather, we focus on the simpler task of domain-incremental learning so as to better highlight the natural temporal evolution of visual concepts. Nonetheless, it is highly likely that other researchers may want to construct class-incremental learning variants of CLEAR benchmark in future, which could be interesting and certainly more challenging.
> * In the past, many published works have tried to define and redefine continual lifelong learning. Like most of them, we share the same point of view that continual lifelong learning focuses on solving a series of non-stationary tasks while aiming to maximize knowledge transfer (both backward and forward); therefore, we attach a thorough list of experimental results in appendix, including those of both the traditional offline protocol that cares about optimizing performance over all timestamps by maximizing accuracy and forward/backward transfer, and our novel online protocol that mostly cares about forward transfer especially to the next-in-line distribution. We do not claim that the online accuracy metric and online evaluation protocol can substitute the offline evaluation protocols, though arguably the locally iid assumption in traditional protocols is unrealistic for many practical applications.
> * Domain adaptation is quite related to continual learning but mostly focuses on transferring from source to target domains with abrupt and clearly defined boundaries (e.g., domain transfer from synthetic to real data), whereas we want to transfer knowledge across a series of smoothly-evolving temporal domains.
>
> > (3) Precise nature of the distribution shift in the proposed dataset is not clear to me.
>
> * Different visual concepts may exhibit different levels of dynamicness: “computer” models certainly changed their visual appearance drastically during the past decades, whereas “hockey” games may have experienced fewer obvious changes. Nonetheless, as social events including sports are mostly temporally dynamic, e.g., hockey events could take place in different locations and feature different teams and themes in different times, such differences are clearly observable from the “hockey” images of CLEAR. Also, sensor changes are obviously undeniable and unavoidable for real world continual visual recognition systems - these are notoriously challenges for next generation autonomous vehicle datasets (e.g., Argoverse 1.0 versus 2.0). Therefore, we agree with the reviewer and believe that it will be an interesting future work to analyze the precise nature of distribution shift in CLEAR or in other visual concepts.

---

> > ### Author Response · Authors · 2021-09-27
> > **Responses to the review (part 2)**
> >
> > > (4) The statement “CLEAR also comes with abundant unlabeled data per time period” seems to be an attempt to make the contribution look bigger than it actually is.
> >
> > * Firstly, we emphasize this point because our experiments demonstrate that unlabeled data is rather helpful for real-world CL - i.e., the delta from adding unlabeled data dwarfs the delta between the best-vs-worst-performing CL algorithm (see Table 2). But most prior CL benchmarks do not make use of unlabeled data. Secondly, acquiring large-scale unlabeled data is still challenging; many such datasets are proprietary due to privacy concerns. CLEAR is able to leverage existing weakly-labeled datasets like YFCC100M.
> >
> > > (5) Table 2 does not have mean + std.
> >
> > * We apologize for omitting mean+std for table 2. For the rest of the experiments in our appendix, we all include 5 runs with different random seeds and show the mean+std. We have updated the manuscript for table 2 with the mean+std.
> >
> > > (6) Replay buffer is too large (one bucket of training images).
> >
> > * We agree that making the replay buffer size smaller would make the benchmark more challenging, and therefore future work could certainly consider using much smaller buffer sizes in order to develop better exemplar maintaining strategies to better capture the temporally moving distributions. Nonetheless, the benchmark is already difficult with such a large buffer size when no unsupervised pre-training is allowed (as in Table 2). We focus on the simplest domain-incremental learning setting with large enough buffer size to emphasize the effectiveness of unsupervised learning and the dynamic nature of the CLEAR benchmark.
> >
> > > (7)  Is YFCC100M not too large to handle in a modern time? Does “merely one day of engineering effort” not mean reasonable effort for dataset construction? How does the performance of algorithms behave with respect to how many data instances per class+bucket are considered (for future extension of CLEAR)?
> >
> > * We disagree with the reviewer and would like to emphasize that YFCC100M is still too large to handle; it is clearly much larger than most datasets in the vision community nowadays -- ImageNet’s 1M labeled images is already one of the largest publicly available datasets and there have been few vision datasets exceeding size of a million. Even though YFCC100M was released 5 years ago, there has been a limited number of works that fully utilize the potential of this weakly labeled image collection. We want to point out that it was not the physical constraint (hard drive is definitely cheap) but rather costly human labor that prevents people from curating datasets and benchmarks out of YFCC100M. If it was not for the CLIP-assisted dataset curation pipeline, our labelling cost would have been scaled up 10-100X (instead of just $5000).
> > * The code that we used for constructing CLEAR only takes one day of “engineering effort”, but this is not the total amount of time used to curate and clean CLEAR. It took at least 400-500 hours of time for the MTurk workers to verify the labeled portion of CLEAR.
> > We focus on the simplest setting with a fixed number of samples and buckets in order to fairly compare all the baselines. Because our dataset curation methodology is flexible, one can extend CLEAR in arbitrary ways, e.g., more classes, more or fewer samples per bucket, more or fewer buckets in total, and etc.
> >
> > > (8)  Is YFCC100M still accessible?
> >
> > * We provide all the steps required to replicate our entire dataset curation process (downloading YFCC100M metadata and raw images/sorting and chunking the data/etc.) in the appendix. In fact the appendix already contains a solution to access YFCC100M data (see appendix page 1’s footnote for the code snippet), which is no longer available on their website but still available on Amazon S3. You may also refer to our github repository (https://github.com/linzhiqiu/continual-learning) for the code we used to download and process YFCC100M data.
> >
> > > (9) “GDumb implies that CL may even be harmful because from-scratch classifiers tend to perform better by ignoring previous experience” is a factually incorrect statement.
> >
> > * We agree that this sentence may cause confusion (as all state-of-the-art CL methods utilize a replay buffer which exactly aims to preserve past experience through external storage). Our purpose is to emphasize that by training a classifier from scratch (i.e., random initialization) for each timestamp (instead of finetuning the previous model checkpoint) already surpasses the performances of most popular CL techniques. We have updated this sentence to “...ignoring learned models from previous timestamps''.

---

> > > ### Author Response · Authors · 2021-09-27
> > > **Responses to the review (part 3)**
> > >
> > > > (10) The way I understand what the authors do, is that they pre-train a model on the same classes as provided in later evaluation, and then check how much this helps when evaluating these classes under some form of later distribution shift.
> > >
> > > * We believe the reviewer has a major misunderstanding about the pre-training step we perform. What we do in this paper is to utilize the unlabeled 0th bucket of CLEAR (with around 0.7M images) to pretrain a MoCo v2 model which serves as a feature extractor for later timestamps. We do not use the “same classes as provided in later evaluation” to pretrain a feature extractor, since we do not have any class labels for the bucket 0th of CLEAR. Clearly though, this additional 0.7M unlabeled images makes the unsupervised pre-training baselines an unfair comparison to the fully-supervised baselines; however, we pointed out that unlabeled data can be readily obtained almost for free from Internet and training a self-supervised model is also no longer costly nowadays. Therefore, we argue that unsupervised pre-training should also be considered as a simple yet effective baseline for future works in CL.
> > >
> > > > (11) Line 215+ states that “offline CL” is equivalent to “unlimited storage” in a sample-storage/size of replay buffer. I do not think this is true. “Offline” evaluation in continual learning just assumes that a “task” can be trained to convergence over the course of multiple epochs (revisiting data) before proceeding to the next “task” (where old task data is then no longer available, in whichever way the task is respectively defined).
> > >
> > > * Thank the reviewer for pointing this out. It is true that unlimited storage is not equivalent to offline CL. We remove this sentence that may cause confusion from our manuscript. Other than this, we believe that our discussion on online v.s. offline CL in Sec 2 (Related work) aligns with the reviewer’s definition.
> > >
> > > > (12) Are you redefining the term “online” for CL?  Why not term it as “open world” instead of “online”?
> > >
> > > * We want to emphasize that we do not redefine the term “online” for CL. What we propose in this paper is a novel online “evaluation protocol” and an “online accuracy metric” for CL. We did mention that our "online" evaluation protocol can still evaluate both offline and online CL algorithms in Sec 2. (Related Works) of our paper.
> > > * We use the term “online” to describe scenarios where real world data comes in an ongoing stream including the test data, and therefore we always train on today’s data and test on tomorrow’s. In this respect, “open world” might be too broad a term to describe our protocol.
> > >
> > > > (13) Table 1 is misleading: (a) mixing datasets with methods on y-axis. (b) A-E are not property of datasets but the way algorithms are constructed and evaluated. (c) The rows and columns are not explained clearly in the text.
> > >
> > > We appreciate your detailed comments on how to improve table 1.
> > > - We are sorry that the citations for “Task-IL”/”Class-IL”/”Dataset Pairs” in the top-row might seem misleading; however, these papers that we cite are well-known papers that adopt each of these CL scenarios when designing their benchmark, and we put these citations there to encourage readers to study those representative examples of CL benchmark scenarios. We do not aim to mix datasets with methods on y-axis, but since there are no existing paper that solely focuses on CL benchmark (most CL papers are a mix of dataset, benchmark, and methods), we try our best to give those representative examples of CL benchmark design in Table 1. We have added this clarification to the caption.
> > > - A-E are indeed not property of datasets but desiderata for CL experiment design, so your understanding that they refer to the ways that the algorithms are evaluated is correct. Experiment design is a property of the benchmark, but not of the dataset, and therefore the checkmarks are referring to whether those existing CL benchmarks have their experiments set up in a way to reflect those desiderata.
> > > - We have a thorough discussion on desiderata A-E in Sec. 2 of appendix. We explain the other strengths of CLEAR (e.g., realistic image distributions) in both main text and table caption. Unfortunately we do not have enough space to explain how each of these individual checkmarks are selected, but we assume most of them are self-explanatory and applied to the majority of the cases (please correct us if you believe any of the checkmarks are falsely labeled, thanks!)
> > >
> > > > (14) A large number of the citations are incomplete, inconsistent or wrong.
> > >
> > > * Could the reviewer elaborate about the incomplete/inconsistent/wrong parts of the citations? We have checked and updated our citations to ensure that papers published in major conferences are not cited in their arxiv versions.

---

> > > > ### Author Response · Authors · 2021-09-27
> > > > **Responses to the review (part 4)**
> > > >
> > > > > (15) The authors should ideally have either uploaded their code already, or at least provided a copy in the supplementary upload of the paper submission.
> > > >
> > > > * In fact we have already included the code in our website: We have one public github repository for dataset curation, and another public github repository for evaluation. These links are also available on our website (https://clear-benchmark.github.io)!
> > > >
> > > > > (16) Question about datasheet: “Does the dataset relate to people?”
> > > >
> > > > * We have updated the response to this question in the datasheet from “No” to “The images may contain the presence of human beings”. We have also corrected the surrounding questions to be more careful in our wording.
> > > >
> > > > > (17) YFFC100M clearly states “All the photos and videos provided in the list are licensed under one of the Creative Commons copyright licenses, and as such they can be used for benchmarking purposes as long as the photographer/videographer is credited for the original creation”. Is this actually being done for each extracted image?
> > > >
> > > > * We include the licenses for each of the individual image objects in CLEAR, which is also released in the form of CSV files to the public on our website (check out the metadata files). The metadata CSV files also contain the photographer information, thereby ensuring the original creators have been credited.
> > > >
> > > > > (18) The authors should ideally host their dataset in a form that has a DOI and supports versioning for future changes and reproducibility.
> > > >
> > > > * We thank the reviewer for the suggestion, and we have uploaded our dataset to Zenodo with a DOI: 10.5281/zenodo.5528798
> > > >
> > > > Finally, we really do appreciate all the harsh comments and constructive feedback. We hope that through this rebuttal we have clarified some of the major misunderstandings and addressed most of the writing issues you pointed out earlier. We hope you would reconsider the scoring of our paper based on the improvements we made since your feedback. We look forward to engage you in more constructive discussion.

---

> > > > > ### Comment · Reviewer_kspM · 2021-09-28
> > > > > **Partially good answers, rating has been raised. Concerns remain with respect to presentation of related work**
> > > > >
> > > > > I sincerely thank the authors for responding in detail to my lengthy review. I have made an update to my initial review above and have raised my rating by two points. (See above, as open review does not seem to notify authors of these changes automatically)
> > > > >
> > > > > Many of the responses given by the authors are appreciated, especially the authors' positioning and additional clarifications with respect to the unsupervised experiments and acknowledgement of some suggested changes.
> > > > >
> > > > > Unfortunately, it is rather hard to only imagine the changes and it would perhaps be required to see a future updated version of the paper to come to a final conclusion of acceptance. Newly added elements of the presently omitted parts in section 2 of the dataset sheet documentation will further require additional review. Once more, I emphasize that the ideas have great potential, but some of the writing and presentation in the current form need major improvements in my humble opinion.
> > > > >
> > > > > For instance, one of the key remaining concerns, is the misleading presentation of related works and benchmark/evaluation setups of table 1. Although the authors seem to acknowledge and agree with the criticism, there does not seem to be a concrete suggestion of how to do a proper revision of this section. To clarify, I do not believe it is sufficient to provide a small note that the table might be misleading and/or add optional statements in the appendix. The story should be self-contained in the main body and present a factual narrative that does not conflate benchmarks, dataset propositions and novel algorithm suggestions.
> > > > >
> > > > > I also still think the wordings of contribution are not on point. It would be appreciated if the authors clarify/revise this. That is, it remains unclear to me why the unsupervised portion of the dataset, which in my understanding is not subject to the CLIP + MTurk process for labelling, is considered a central contribution if it essentially boils down to adopting a subset from an existing dataset?

---

> > > > > > ### Author Response · Authors · 2021-09-29
> > > > > > **Follow-up Response to Reviewer # kspM [paper updated]**
> > > > > >
> > > > > > Dear Reviewer,
> > > > > >
> > > > > > Thank you for updating your rating and many suggestions on improving our paper! We have uploaded a new version in response to your comments. We summarize key changes below:
> > > > > >
> > > > > > > Revising Table 1
> > > > > > - We have made significant updates to Table 1. First, a bit of clarification on the original table. Our paper's focus was mainly on the last set of rows (last 3 rows) that describes crucial properties of our data (real-world, smooth temporal evolution, and continual un/semi-supervised). The other rows in the old Table 1 were an attempt to situate our benchmark evaluation with respect to the guidelines provided by prior work: “CL problem scenarios” (incremental domain vs incremental classes vs incremental tasks) were outlined in [1] and the “algorithmic desiderata for CL evaluations” was outlined in [2]. Note that we don’t necessarily advocate for these prior taxonomies, but included their categorization to connect to the prior literature. However, as per your valuable comments, we made two significant changes to Table 1:
> > > > > >     1. We have revised Table 1 to focus exclusively on dataset properties in the rows and changed columns to refer to prior CL dataset.
> > > > > >     2. We moved the discussion on "problem scenarios" and "evaluation desiderata" which were originally in Table 1 to the in-line text (in lines 205-209 & 065-070), described below.
> > > > > >
> > > > > > - In terms of CL problem scenarios [1], our existing benchmark falls squarely into incremental domain learning, though one could pursue incremental class learning in the future (which we hope to do!). In terms of evaluation desiderata, [2] outlines a series of experimental design choices to discourage “trivial” solutions, such as training separate models for separate tasks. Note that these desiderata impose restrictions on the algorithm, such as how classifier heads are shared across tasks. We agree with the reviewer that traditionally, benchmarks and evaluation criteria would be agnostic to such algorithmic choices. Farquhar and Gal [2] go so far as to claim that “No prior-focused continual learning system has so far been shown to succeed when all five desiderata are applied.” We point out that our evaluations -- combining our dataset, metrics, and baseline algorithms -- do satisfy these criteria. We have revised the text accordingly.
> > > > > >
> > > > > > > Regarding data sheet
> > > > > > - We include a revised datasheet description in the supplemental materials.
> > > > > >
> > > > > > > Unlabeled data
> > > > > > - Finally, in terms of positioning with respect to unlabeled data, the reviewer is quite correct in pointing out that unlabeled data has not been verified via MTurk (due to the exorbitant costs for doing so). But our results definitely show that unlabeled data is crucial for strong performance, and no prior CL dataset includes them (as Table 1 now clearly shows). In principle, existing CL datasets could be repurposed for semi-supervised learning by ignoring labels, but self-supervised representation learning tends to require large scale data not present in current supervised CL datasets [3].
> > > > > >
> > > > > > > "wordings of contribution are not on point... it remains unclear to me why the unsupervised portion of the dataset, which in my understanding is not subject to the CLIP + MTurk process for labelling, is considered a central contribution"
> > > > > > - As per your feedback, we looked back at the text in our paper and found all of them to be factually correct (see 80-82, 97-99, 354-355). In particular, all of these lines are merely stating a property of CLEAR that unsupervised pretraining helps, and not stating that unlabeled data is a contribution. Please let us know if we missed any lines which say that unsupervised data is a contribution and we would be happy to edit the paper asap.
> > > > > >
> > > > > >
> > > > > > Finally, we hope that we have addressed your concerns. Please let us know if any of your concerns are remaining. We are happy to revise any specific text you feel is out-of-line. We look forward to more of your suggestions and reconsideration of the rating!
> > > > > >
> > > > > > ### References
> > > > > >
> > > > > > [1] Re-evaluating Continual Learning Scenarios: A Categorization and Case for Strong Baselines, Yen-Chang Hsu, Yen-Cheng Liu, Anita Ramasamy, Zsolt Kira, NeurIPS Continual LEarning Workshop 2019. [**cited 119 times**]
> > > > > >
> > > > > > [2] Towards Robust Evaluations of Continual Learning. Sebastian Farquhar, Yarin Gal. Neurips Continual Learninig Workshop 2018. [**cited 118 times**]
> > > > > >
> > > > > > [3] Momentum Contrast for Unsupervised Visual Representation Learning. Kaiming He, Haoqi Fan, Yuxin Wu, Saining Xie, Ross Girshick. CVPR 2020. [**cited 1662 times**]

---

> > > > > > > ### Comment · Reviewer_kspM · 2021-10-04
> > > > > > > **Pdf update is appreciated**
> > > > > > >
> > > > > > > I again thank the reviewers for engaging with my feedback and uploading a revised version of the pdf.
> > > > > > >
> > > > > > > With more such revisions, I believe the paper will be shaped into a very useful contribution. Right now, I am left with the impression that the authors require some more time to go through all the reviewers' feedback.
> > > > > > >
> > > > > > > To give two short examples:
> > > > > > >
> > > > > > > * Whereas the modifications to the discussion and table 1 are already appreciated, I would recommend to think this through fully. The changes feel somewhat rushed. The table now only has three items. One of them is whether a dataset (on the column) is "semi/unsupervised" (on the row). I fully understand what the authors write in the rebuttal about the prevalent way CL literature uses benchmarks. The response makes their intention clear. However, I nevertheless find the form of presentation misleading, as datasets can be used in super/unsupervised fashions purely based on choice. We can always choose to simply not use labels. This is very typical in the literature, e.g. on semi-supervised ImageNet, few-shot learning, active learning literature etc. It would be best to just revise this section properly to get the point across, without having such misleading standalone tables.
> > > > > > > * I agree with the authors rebuttal on none of the contributions being factually incorrect. At the same time, in my humble opinion, some of the statements are easily misread to conflate the contribution. For example: "Besides high-quality labeled data, CLEAR also comes with abundant (0.7M) unlabeled domain data per time period". These contribution statements to me are still conflating the actual paper contributions with incremental things. As the authors have written in their rebuttal, unsupervised data points do not undergo their proposed pipeline, so essentially this data to me just looks like a straightforward set taken from YFCCM.
> > > > > > >
> > > > > > > Similar points can be made about the changes to the dataset documentation. While it is commendable that the authors immediately attempted to address the gaps for previously blank aspects, this should be done with more than just adding single sentences in various places. A proper discussion would be useful. However, I would also not necessarily fundamentally object, if the area chair thinks it is okay to accept this paper on the condition that further additions/revisions are made for the camera ready.

---

> > > > > > > > ### Author Response · Authors · 2021-10-06
> > > > > > > > **Response**
> > > > > > > >
> > > > > > > > We thank the reviewer for their continued engagement. The primary remaining concern seems to be positioning of semi-supervised labels; from the reviewer’s perspective, semi-supervision shouldn’t be regarded as a property of a dataset because one can always simulate semi-supervision by ignoring labels from an existing supervised dataset. But crucially, our semi-supervised data is “in-the-wild” and not curated. It is widely accepted that self-supervision on in-the-wild data is far more realistic and challenging, since curated data often make use of labels during dataset construction. This is why, for example, self-supervision experiments on “unlabeled Imagenet” can be met with a degree of skepticism [1].  In fact, [1] proposes a benchmark for (non-continual) self-supervision that makes use of YFCC100M, much like us! Our experiments demonstrate that adding in-the-wild self-supervised data improves baseline performance by 10-20%, which dwarfs the improvement due to virtually every algorithmic innovation proposed in the continual learning literature; how is this is “incremental”? The contribution of a benchmark is not just the data itself, but the collective combination of data, metrics, and baseline experiments that lay out the research landscape of future issues that need to addressed.
> > > > > > > >
> > > > > > > > [1] Scaling and Benchmarking Self-Supervised Visual Representation Learning, CVPR19 [174 citations]

---

> > > > > > > > > ### Comment · Reviewer_kspM · 2021-10-06
> > > > > > > > > **Thank you for the additional response**
> > > > > > > > >
> > > > > > > > > It is good to see that the authors continue to engage with the reviews.
> > > > > > > > >
> > > > > > > > > I agree with the authors' statements on a benchmark being composed of more than the data itself. I do concur that the paper provides a general valuable contribution to the continual learning community.
> > > > > > > > >
> > > > > > > > > My main strong concerns have already been addressed in a pdf revision, the remaining concerns were primarily about the presentation format and language. I do acknowledge that some of these may be subjective, but I can also see that the authors take into account feedback.
> > > > > > > > >
> > > > > > > > > I believe it is thus reasonable to assume that the authors will continue their text improvements up until the camera ready deadline. I'm thus inclined to vote for paper acceptance. (Note that I can no longer edit my original review box/rating because open review seems to have ended the discussion period officially already).
> > > > > > > > >
> > > > > > > > > Ultimately, I think the introduction of this benchmark will outweigh any potential remaining pitfalls.

---

### Official Review · Reviewer_UqHL · 2021-09-20

**Rating:** 5
**Confidence:** 3

**Strengths:**

(1) The proposed benchmark considered the realistic temporal evolution of images and their concepts in continual learning, which I think can better capture the dynamic nature in real scenarios.

(2) The combination of using CLIP for coarse-level data annotation and relying on humans for a fine-grained curation looks interesting. I agree that this can improve the scalability of data collection.

(3) An alternative online evaluation protocol has been developed for better capturing the domain shift, by always using today’s data to train and tomorrow’s data to test.


**Weaknesses:**

(1) The gap between offline accuracy and online accuracy seems to be quite marginal for different baseline methods. For example, in Table 2, LwF [27] achieves 76.4% offline accuracy and 77.3% online accuracy. Therefore, the proposed continual learning benchmark is not more challenging than the offline evaluation protocol (i..e., on the i.i.d. data).

(2) Unsupervised pretraining on YFCC-S0 (0.7M images of segment 0) achieves a 90%+ online accuracy. Wouldn’t it mean the proposed benchmark is too simple and there is not much room for improvement? Why do we still need the proposed benchmark for continual learning? Or does it mean unsupervised learning is the most promising solution to continual learning problems?

(3) Similar to CIFAR-10, the dataset also has 10 class labels, which dynamically evolve over time. Not sure what rationale behind this choice of labels, but it seems to me that several class labels are not exclusive to each other. For example, “cosplay” and “dress” could co-occur in the same images. Also, some labels are more closely related concepts than others, such as “dress” and “sweater” vs. “hockey” and “baseball” vs. “computer” and “camera”.

(4) How many labeled images are left after human curation?





================

Post-rebuttal update:

I thank the authors for providing very detailed and thoughtful feedback. Some of my concerns have been clarified, but I still feel that more efforts have to be done to make this dataset more challenging for both CL and self-supervised methods. Therefore, I raised my rating from 4 to 5.

**Additional Feedback:**

N/A

**Clarity:**

The paper is overall well written. Typo: In Figure 2, “visio-” => “visuo-”.


**Correctness:**

The dataset is constructed in a sound way, though there are some unclear issues I mentioned in the above. The evaluation methods and experiment design are appropriate and performed correctly.


**Documentation:**

The authors have included detailed documentation on data collection and organization, availability and maintenance and ethical and responsible use. I think there is also sufficient detail to support reproducibility.


**Relation To Prior Work:**

The prior continual learning benchmark and baseline methods have been clearly discussed in the Related Work section.


**Summary And Contributions:**

This work proposed a new continual learning benchmark, named CLEAR, from the publicly available image collection YFCC100M. The proposed benchmark considered the temporal evolution of visual concepts over time from 2004 to 2014, and the data construction process is scalable and semi-automatic by using the pre-trained CLIP for label annotation and then human annotators for curations. Finally, several baseline methods and analysis were performed to show the necessity of the proposed benchmark.

---

> ### Author Response · Authors · 2021-09-27
> **Responses to the review**
>
> We are glad that the reviewer appreciated our efforts in constructing a more realistic benchmark for continual learning and hope our novel dataset curation methodology can benefit future large scale vision dataset and benchmark curation.
>
> Responses to the weaknesses:
>
> > (1) “The proposed evaluation protocol and metrics are not more challenging than the offline counterpart?”
>
> Online accuracy (as well as the online evaluation protocol) was not proposed because it is more challenging, but because it was more (in our experience) realistic since it does not rely on assumptions of locally iid data. This is because gathering data, annotating labels, training and deploying ML models all take a considerable amount of time; once the model is deployed, the real world data distribution has most likely evolved. Online accuracy and evaluation protocol are proposed to address this gap by testing a model on the immediate future task/distribution. As we consistently observe the gap between offline and online accuracy across all baseline CL methods, we simply cannot deny the existence of this gap in CLEAR and most likely in other more complicated real world application scenarios. Future research on CL may adopt either the traditional offline protocol or our novel online protocol depending on whether the locally iid assumption holds.
>
> > (2) Does the superior performance of the unsupervised pre-training baselines means that CLEAR is already saturated without further algorithmic improvement?”
>
> We agree that unsupervised pre-training seems to be the most promising direction for future CL baselines, benefiting from tremendous advances in self-supervised learning (MoCo, SimCLR, BYOL, etc.) in recent years. That said, fully-supervised CL baselines (SOTA algorithms including GDumb, A-GEM, etc.) largely fall short on CLEAR and only achieve around 70% accuracy despite having a large buffer size of an entire bucket of training images, which is far from saturated (unlike other small scale CL benchmarks such as Permuted-/Split-MNIST). Unsupervised pre-training baselines on the other hand achieve much better performance and it might seem saturated at first glance. However, we want to point out that as the first CL benchmark based on CLEAR dataset, we do not aim to make the problem artificially challenging and we (1) only focus on the least challenging domain-incremental setting (in contrast to class-increment setting for which CLEAR can also be adapted), (2) we use a large replay buffer size (around 3K images) of an entire training bucket of labeled images, whereas the other benchmarks such as Permuted-MNIST have a much smaller buffer size (around several hundreds). Future benchmarks based on the CLEAR dataset could try to either reduce the gap between fully-supervised and unsupervised pre-trained CL baselines or control the size of the replay buffer.
>
> > (3) What is the rationale behind the choices of classes in CLEAR? “Cosplay” and “dress” seem to overlap with each other?
>
> Thanks for the careful review of the choices of classes in CLEAR. We pick the classes based on whether they are temporally dynamic or not, e.g., social events, clothing styles, and technology products, all of which are naturally changing over time. We pick a few classes from each of the dynamic super categories as listed in Sec. 3 “concept selection”. We do realize there might be some overlap between the classes, which is very common for real world recognition problems (ImageNet also has the same issue [1]). Therefore, we carefully design the labelling policy to avoid confusion for MTurk workers. For example, in the appendix we provide specific items that may appear in those classes. In the current design of CLEAR, cosplayers wearing dresses belong to the “cosplay” but not the “dress” class. The “dress” class refers to common dress that people wear daily (e.g., skirt, ballgown, etc.), whereas “cosplay” classes refer to special social events in which people are dressed up as characters from anime/movies.
>
> > (4) How many (labeled) images are left after human curation?
>
> For each visual concept per bucket, we retrieve the top-scoring 600 images; after MTurk label verification, we select the top-scoring 300 images from the remaining verified images. These details of dataset curation are available in Sec.1 of appendix.
>
> [1]  From ImageNet to Image Classification: Contextualizing Progress on Benchmarks. Dimitris Tsipras, Shibani Santurkar, Logan Engstrom, Andrew Ilyas, Aleksander Madry.

---

### Official Review · Reviewer_84Jk · 2021-09-20
**interesting new benchmark for continual learning**

**Rating:** 7
**Confidence:** 3
**Correctness:** No red flags.
**Clarity:** Yes.

**Strengths:**

[see review]

**Weaknesses:**

[see review]

**Additional Feedback:**

N/A

**Documentation:**

Yes.

**Ethics:**

Not much that I can think of.

**Relation To Prior Work:**

Yes.

**Summary And Contributions:**

This work introduces a new benchmark for continual learning. It compellingly describes many of the existing benchmarks as insufficient (due to either being artificial or not truly aligning with the premise of continual learning).

In general, the proposed method for collecting the data, filtering etc seems reasonable. The authors gave obviously given a lot of thought to making a good dataset. I am concerned by the number of "moving parts": the choices of the dynamic visual concepts, the chunking into 10 parts, the usage of CLIP and finally mturk. Each of these steps has its own set of design choices and hyper-parameters and it's not clear just how robust the whole procedure is to minor changes in each of these steps (meaning, would we get a similar kind of data?). I wonder if the reservoir sampling details in Section 5 are also a bit overfit to CLEAR, but that is a minor issue.

All in all, I like the benchmark and I think it will be useful to the community. It has a more real world nature to it so likely models evaluated on it will have more practical value.

---

> ### Author Response · Authors · 2021-09-27
> **Thank you for the review!**
>
> We are glad that the reviewer agrees that current CL benchmarks are insufficient and we believe CLEAR is one of the first attempts towards a more realistic CL benchmark for computer vision.
>
> > (1) “I am concerned by the number of "moving parts": the choices of the dynamic visual concepts, the chunking into 10 parts, the usage of CLIP and finally mturk. Each of these steps has its own set of design choices and hyper-parameters and it's not clear just how robust the whole procedure is to minor changes in each of these steps (meaning, would we get a similar kind of data?).”
>
> There are indeed a number of “moving parts” in defining our dataset, but we believe this to be the case for any large-scale data curation effort.  Our intention in formalizing our curation process was to make it structured and repeatable. While it is difficult to provide precise quantitative metrics about the stability of our pipeline, we outline the major design choices below.
>
> * **Choice of concepts**: We choose visual concepts that are conceivably dynamic, e.g. social events and technology products are dynamic with respect to time. There are potentially more visual concepts that are dynamic, e.g. phone, shirts, and etc. It would be an interesting future work to gather other CL datasets featuring different dynamic visual concepts (potentially from super-categories other than the 3 we listed in paper) and to compare them with CLEAR.
> * **Choice of temporal resolution of bucket**: We find it fairly straightforward to produce fewer/more buckets with different temporal resolutions (months or quarterly buckets versus yearly buckets).
> * **Auto-curation with CLIP**:  We use CLIP in order to reduce the data cleaning cost by orders of magnitude, and thus it is definitely an indispensable part of our pipeline.
> * **Human verification**: We believe that our final step of MTurk verification can be replaced by other human label verification tools or services.

---

### Official Review · Reviewer_eTGx · 2021-09-21
**Good paper**

**Rating:** 6
**Confidence:** 2
**Correctness:** Yes
**Clarity:** Yes, it is well written.

**Strengths:**

The major strengths of CLEAR over prior CL benchmarks include (1) smooth and realistic temporal evolution of visual concepts with real-world imagery, enabling a more practical "online" (i.e., train on past, test on future) evaluation protocol (2) high-quality labeled data along with abundant unlabeled samples per time period for continual semi-supervised and unsupervised learning.

**Weaknesses:**

The author only compare linear model with ResNet18. Is it enough for this task?

**Additional Feedback:**

Explain the above weakness.

**Documentation:**

Yes.

**Ethics:**

No.

**Relation To Prior Work:**

Yes.

**Summary And Contributions:**

1.The authors build CLEAR from existing image collection (YFCC100M) by proposing a novel low-cost visio-linguistic dataset curation approach.
2.The authors find that state-of-the-arts CL algorithms that only utilize fully-supervised data fall short whereas unsupervised pretraining provides significant boost.
3. The authors introduce a biased reservoir-sampling algorithm that dynamically caches more recent training data, achieving the new state-of-the-arts while still leaving large room for improvement.

---

> ### Author Response · Authors · 2021-09-27
> **Response**
>
> We thank the reviewer for acknowledging the strengths of CLEAR over prior CL benchmarks.
>
> > (1) “The author only compare linear model with ResNet18. Is it enough for this task?”
>
> We want to point out that we did a thorough list of baseline experiments (certainly not limited to naive linear models and ResNet18) in both main text and appendix: (1) We evaluate classic state-of-the-art fully-supervised CL methods (A-GEM, LwF, GDumb, and etc.) using ResNet18 as the backbone network. (2) We evaluate both linear models and non-linear models (specifically MLPs, see appendix) over features extracted unsupervised pre-trained models (MoCo v2) trained on the very first temporal bucket (of 0.8M unlabeled images) of CLEAR. The appendix includes the results and visualization for all those experiments (including mean and std for 5 experiments with different random seeds), and we hope our findings could point out more promising directions to develop effective CL techniques, e.g., embracing unlabeled domain data for continual unsupervised/semi-supervised learning.

---

### Decision · Program_Chairs · 2021-10-09

**Decision:**

Accept

**Comment:**

The paper proposes CLEAR - an image recognition benchmark for continual learning based on YFCC100M dataset. The reviewers agree that this benchmark departs significantly from existing datasets in this space in terms of task definition and scale. The scale is achieved by relying on recent high-performance vision-language models (CLIP) and crowd-source effort to ensure quality and deal with certain ethical aspects (e.g. remove inappropriate images). To have a more natural CL setting, the benchmark task is built around the temporal evolution of visual concepts (e.g. computer, dress, etc) over a decade. Since the proposed benchmark is a curated subset of YFCC100M, the authors recommend (and show the benefit of) using the remaining part of YFCC100M for semi-supervised and unsupervised learning, highlighting the shortcomings of current fully-supervised methods for CL.

Despite some initial disagreements around terminology and claimed contributions, the reviewers and the authors engaged in a very constructive review process, resulting in a significant improvement of the submission. After detailed discussion among the reviewers and the area chairs, we concluded that this paper makes a significant contribution to the CL field and has high potential to drive research in the field. There are still some concerns about some of the scores being close to saturation and clarity of describing the contributions, and we strongly encourage the authors to revisit these aspects for the camera ready version.